# Down regulation of vestibular balance stabilizing mechanisms to enable transition between motor states

**Romain Tisserand[1], Christopher J Dakin[2]\*, Machiel HF Van der Loos[3,4], Elizabeth A Croft[5], Timothy J Inglis[1,6,7], Jean-Sébastien Blouin[1,4,7]**

[1]School of Kinesiology, University of British Columbia, Vancouver, Canada; [2]Department of Kinesiology and Health Science, Utah State University, Logan, United States; [3]Department of Mechanical Engineering, University of British Columbia, Vancouver, Canada; [4]Institute for Computing, Information and Cognitive Systems, University of British Columbia, Vancouver, Canada; [5]Department of Engineering, Monash University, Clayton, Australia; [6]International Collaboration on Repair Discoveries, University of British Columbia, Vancouver, Canada; [7]Djavad Mowafaghian Centre for Brain Health, University of British Columbia, Vancouver, Canada

**Abstract** The neural control of transition between posture and movement encompasses the regulation of reflex-stabilizing mechanisms to enable motion. Optimal feedback theory suggests that such transitions require the disengagement of one motor control policy before the implementation of another. To test this possibility, we investigated the continuity of the vestibular control of balance during transitions between quiet standing and locomotion and between two standing postures. Healthy subjects initiated and terminated locomotion or shifted the distribution of their weight between their feet, while exposed to electrical vestibular stimuli (EVS). The relationship between EVS and ground reaction forces was quantified using time-frequency analyses. Discontinuities corresponding to null coherence periods were observed preceding the onset of movement initiation and during the step preceding locomotion termination. These results show humans interrupt the vestibular balance stabilizing mechanisms to transition between motor states, suggesting a discrete change between motor control policies, as predicted by optimal feedback theory.

DOI: https://doi.org/10.7554/eLife.36123.001

\*For correspondence:
chris.dakin@usu.edu

**Competing interests:** The authors declare that no competing interests exist.

## Introduction

The initiation of movement from a stable posture is a central issue in neuroscience because of the need to overcome reflex-stabilizing mechanisms to enable motion (*von Holst and Mittelstaedt, 1950*). Navigating our world involves frequent transitions between posture (e.g. quiet stance) and movement (e.g. locomotion), representing an ideal model to explore this issue. Indeed, balance correcting responses may hinder locomotor initiation and the transitions that are essential to safely navigate our environment (*Ostry and Feldman, 2003*).

Optimal feedback control theory (OFC) (*Todorov and Jordan, 2002*; *Scott, 2004*), a prominent motor control theory, proposes that the brain implements the optimal means of performing a task by adjusting the appropriate feedback gains to reach the goal. Specific postural configurations and movement patterns are considered separate control policies, one of which must be disengaged (i.e. by decreasing sensory feedback gains) prior to the engagement of another (i.e. by adjusting and

**eLife digest** Crossing Abbey Road is something of a paradox in neuroscientific terms. As you stand waiting to cross, tiny movements of your body – such as those due to breathing – cause you to sway by small amounts. To prevent you from falling over, your brain makes active corrections to your posture. These posture-correcting mechanisms oppose movements such as sway and keep you standing upright. But what happens when you want to cross the road?

To get you moving, your brain has two options. It could temporarily suppress the posture-correcting mechanisms. Or it could reconfigure them so that they work in a different way. The posture-correcting mechanisms rely upon sensory input from various sources. These include the vestibular system of the inner ear. The vestibular system tells the brain about the position and movement of the head in space and relative to gravity. Monitoring vestibular system activity as a person starts to move should thus reveal what is happening to the posture-correcting mechanisms.

Tisserand et al. asked healthy volunteers to transition between standing still and walking, or to shift their weight from one foot to the other. At the same time, small non-painful electric currents were applied to the bones behind the volunteers' ears. These currents induced small changes in vestibular system activity. Sensors in the floor measured the forces the volunteers generated while standing or walking, thereby revealing how they adjusted their balance. The results showed that the brain suppresses its posture-correcting mechanisms before people start or stop moving.

These findings have implications for robotics. They could make it easier to program robots to show smooth transitions into and out of movement. The findings are also relevant to movement disorders such as Parkinson's disease. One common symptom of this disorder is freezing of gait, in which patients suddenly feel as though their feet are glued to the ground. Understanding how the brain controls movement transitions may reveal how such symptoms arise.

DOI: https://doi.org/10.7554/eLife.36123.002

increasing sensory feedback gains) (*Cluff and Scott, 2016*). In this respect, OFC seems to diverge from other motor control theories, such as referent (threshold) control theory (*Asatryan and Feldman, 1965*; *Ostry and Feldman, 2003*). This latter theory suggests that a transition between postures involves a monotonic shift in the referent body orientation (*Feldman et al., 2011*; *Mullick et al., 2018*), transforming posture-stabilizing mechanisms into movement-inducing ones. To determine whether transition between motor states is comprised of discrete events or is a continuous process, we examined the temporal dynamics of sensory feedback gains prior to and during the transition between two motor states. Specifically, we examined the contribution of vestibular sensory signals to the control of balance during transition between quiet standing and locomotion as well as between two standing postures.

The vestibular system encodes motion of the head in space, and is vital to our ability to maintain stability during both quiet standing and locomotion (*Angelaki and Cullen, 2008*; *Goldberg et al., 2012*). Vestibular signals influence the activation of muscles engaged in the control of posture. This influence depends on the muscles' active engagement in balance control, and the alignment of the muscles' mechanical action with both the plane of instability and the direction of the vestibular disturbance (*Lund and Broberg, 1983*; *Britton et al., 1993*; *Fitzpatrick et al., 1994*; *Luu et al., 2012*; *Mian and Day, 2014*; *Forbes et al., 2016*). Contextually-dependent vestibular responses have been observed during both quiet standing (*Lund and Broberg, 1983*; *Fitzpatrick et al., 1994*; *Marsden et al., 2002*; *Son et al., 2008*; *Luu et al., 2012*; *Mian and Day, 2014*; *Forbes et al., 2016*) and locomotion, where the magnitude of vestibular-evoked muscle responses are modulated further during the phase of the gait cycle (*Orlovsky, 1972*; *Matsuyama and Drew, 2000*; *Blouin et al., 2011*; *Dakin et al., 2013*; *Forbes et al., 2017*). The vestibular control of balance during locomotor or posture-to-posture transitions can be monitored by continuously quantifying the magnitude of vestibular-evoked balance responses over the transition. A similar approach has been used previously to monitor the time course of engagement and disengagement of the vestibular control of balance between self- and externally-driven states of balance control (*Luu et al., 2012*).

Here, the continuity of the vestibular influence on postural control during transitions was assessed to determine whether a discrete change in balance correcting mechanisms accompanies transitions

between two motor states. We hypothesized that a discrete suspension in the vestibular control of balance would occur prior to the transition, due to the disengagement of the current control policy prior to the implementation of the next control policy, as predicted under OFC theory.

## Results

In the first study, ten healthy young males completed two blocks of 105 trials where they were instructed to stand quietly, then initiate locomotion, walking at their preferred speed for ~3.5 m, and then to turn around and walk back to their initial position where they were instructed to stand quietly again (defined as the 'locomotor transition' experiment). During each trial, subjects performed two transitions: once at the initiation of locomotion and again at the termination of locomotion. Participants performed one block with the head facing forward and another block with the head turned 90° over the left shoulder (*Figure 1*). Participants received 30 s of a non-painful electrical vestibular stimulus (EVS) behind the ears (Materials and methods) which lasted the duration of each trial. Forceplates were imbedded in the floor on which participants stood and walked, in order to record the ground reaction forces (GRF) generated during the trials. The GRF were used to identify the different phases of movement: quiet standing, transition and walking (*Figure 1*). The vestibular contribution to balance control was quantified using a time-frequency coherence analysis (*Blouin et al., 2011*) between the EVS and GRF signals for each of the three phases of the movement (Materials and methods). Coherence analysis measures the linear relationship between the EVS and GRF signals and is bounded between 0 and 1. In the second study, six additional subjects (young adults, three females) completed one block of 105 trials where they were instructed to stand quietly, then shift their body-weight laterally to redistribute their weight so that 90% was supported by their preferred leg and only 10% by their non-preferred leg (defined as the 'posture-to-posture transition' experiment). Once participants redistributed their body-weight, they were asked to maintain this uneven weight distribution. In this task, participants received EVS for the duration of each trial (15 s).

### A vestibulo-motor null period preceding the transition from quiet standing to locomotion

During the quiet standing period preceding the onset of locomotion, subjects swayed, generating low amplitude shear forces at the feet (root-mean-square of 0.39 ± 0.3 N) and relatively small accelerations of the head (averaged horizontal linear acceleration <0.01 ± 0.3 m/s²) (*Figure 2*). All subjects (n = 10) also exhibited significant EVS-GRF coherence across the 0–10 Hz bandwidth in both head orientations (*Figure 2*). On average, coherence peaked with a magnitude of 0.29 ± 0.11 and 0.41 ± 0.07 at frequencies of 2.1 ± 0.6 and 2.4 ± 0.9 Hz, for the head forward and the head left conditions respectively (*Figure 3*).

At the end of the quiet standing period, that is immediately preceding the onset of the transition period, the forces applied to the body, the linear acceleration of the head and the angular velocity of the head all remained in the range observed during the quiet standing period (*Figure 2*) indicating the initiation of locomotion had yet to begin. However, during this period, the EVS-GRF coherence decreased in all subjects (n = 10). Coherence fell below the 99% confidence limit for both head orientations: 0.435 ± 0.191 s prior to the onset of the transition, for a duration of 0.860 ± 0.260 s with the head facing forward (*Figure 4*) and 0.259 ± 0.143 s before the onset of the transition when the head was turned to the left. However, unlike the head forward condition, coherence did not return once subjects began to walk with the head turned to the left (*Figure 2 and 4*).

During the transition period, EVS-GRF coherence returned in the head forward condition (*Figures 2* and *4*). It peaked on average with a magnitude of 0.18 ± 0.05 at 6.6 ± 2.0 Hz across subjects (n = 10, *Figure 3*). Periods of significant EVS-GRF coherence continued during locomotion (i.e. the two first steps) for the head forward condition, but only during the double-support phases (shaded grey areas, *Figure 2*). The maximum coherence measured during the first step of locomotion (*Figure 1*) in the head forward condition was 0.15 ± 0.06 at 3.9 ± 1.9 Hz (n = 10, *Figure 3*).

In general, EVS-GRF coherence in the head forward condition was greater during quiet standing than during both the transition and the first step (*Figure 3*). Peak coherence during quiet standing was significantly greater than during the transition (difference of (Δ)=0.17 ± 0.10, $t_{(9)}$ = 3.32, p=0.009) and the first step (Δ = 0.14 ± 0.11, $t_{(9)}$ = 3.12, p=0.012) (*Table 1*). Peak coherence during

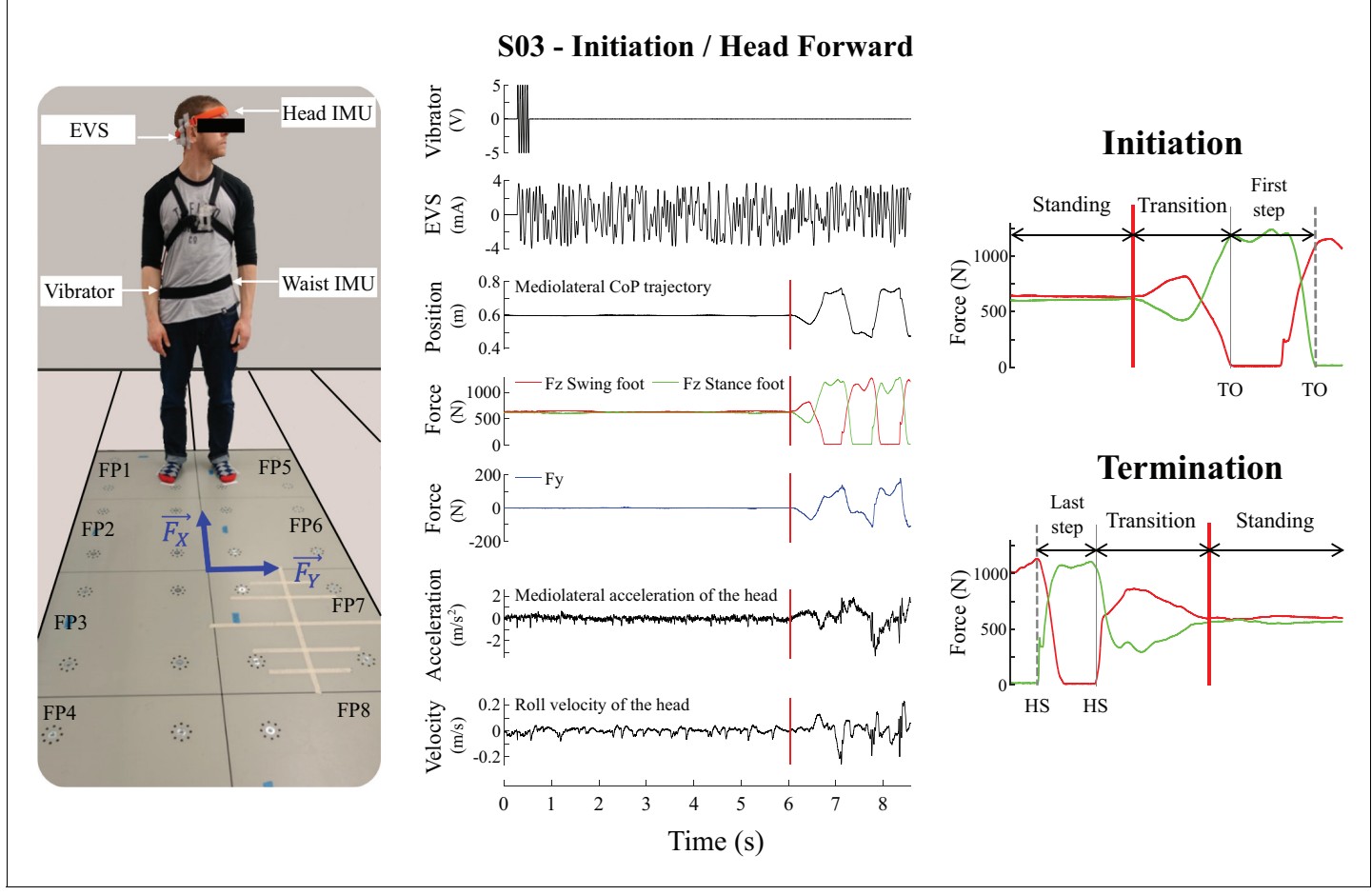

**Figure 1.** Experimental set-up and the signals recorded during the experiment used to detect the movement phases. The left panel is a subject in the initial posture before locomotion initiation in the head left condition. The subject is standing on the forceplates and equipped with the electrodes on his mastoid processes, inertial measurement units (IMU, on the head and the waist) and vibrator, the laser mounted on the orange headband. $F_x$ and $F_y$ correspond to the shear forces measured in the anteroposterior and mediolateral directions, respectively. The middle panel illustrates single trial data from one representative subject during locomotion initiation in the head forward condition. From top to bottom: input signal to the vibrator, electrical vestibular stimulation (EVS) signal, mediolateral centre of pressure trajectory, vertical reaction forces ($F_Z$), shear force in the direction of the vestibular-induced perturbation ($F_Y$), linear acceleration of the head in the direction of the vestibular-induced perturbation, and roll angular velocity. The vertical red line is the onset of the transition period. The right panel shows the vertical force signals at initiation (top) and termination (bottom) used to define movement phases for statistical analysis: quiet standing (between first/last instant and vertical red line), transition (between vertical red line and vertical grey line), and first/last step (between vertical grey line and vertical dashed grey line). Abbreviations used: TO = toe-off; HS = heel-strike.

DOI: https://doi.org/10.7554/eLife.36123.003

quiet standing also occurred at a lower frequency than during the transition ($\Delta = -4.4 \pm 1.9$ Hz, $t_{(9)} = -7.50$, p<0.001) and the first step ($\Delta = -4.8 \pm 1.9$ Hz, $t_{(9)} = -2.98$, p=0.015) (*Figure 3*). No statistical difference was observed between the transition and the first step (p=0.073).

## A vestibulo-motor null period preceding the transition from locomotion to quiet standing

During the locomotion period preceding locomotor termination (i.e. the last two steps), all subjects (n = 10) exhibited significant EVS-GRF coherence when the head was facing forward but not when the head was turned to the left (*Figure 5*). EVS-GRF coherence showed a peak magnitude of $0.14 \pm 0.03$ at $4.6 \pm 2.7$ Hz during locomotion for the head forward condition (*Figure 3*). As we observed significant coherence for every double-support phase following the transition at locomotion initiation in the head forward condition (shaded grey areas, *Figure 2*), we expected similar significant coherence levels in the double-support phase prior to the transition associated with the

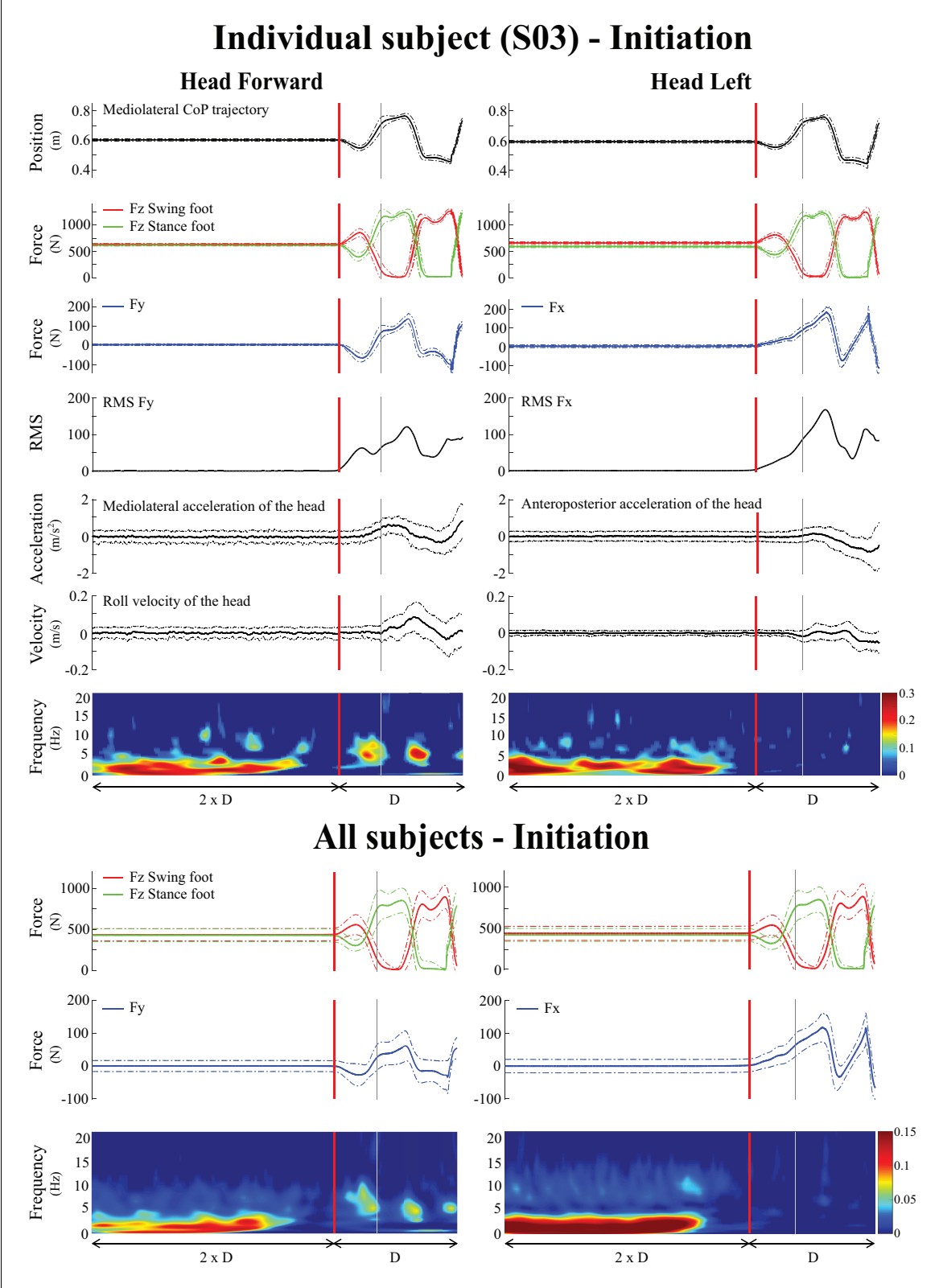

**Figure 2.** Averaged behavioral and time-frequency coherence results for one representative subject (n = 1, top) and across all subjects (n = 10, bottom) during locomotion initiation. Time τ was normalized based on the duration (D) measured between transition onset and the third toe-off. In both lower and upper panels, the left panel is the head forward condition and the right panel is the head left condition. As in Figure 1, the vertical red line shows the transition onset and the vertical grey line shows the transition end. Individual results (top) represent the average ± one standard deviation (n = 100

*Figure 2 continued on next page*

*Figure 2 continued*

trials) of (from top to bottom): mediolateral centre of pressure, vertical forces ($F_Z$), shear force in the direction of the vestibular-induced perturbation ($F_Y$ = mediolateral; $F_X$ = anteroposterior), root mean square (RMS) of that same shear force, horizontal acceleration in the direction of the vestibular-induced perturbation, angular roll velocity of the head, and time-frequency coherence. The pooled results (bottom), represent the average ± one standard deviation (n = 1000 trials) of (from top to bottom): vertical forces ($F_Z$), shear force in the direction of the vestibular-induced perturbation and time-frequency coherence. Solid lines represent the average value while the dashed line is one standard deviation. The grey-shaded areas represent the double support periods during locomotion, in the head forward condition only. In the time-frequency coherence graphs, all non-significant coherence values have been set to zero based on a 99% confidence limit (single subject = 0.045, pooled data = 0.005). For illustrative purposes, vertical forces of the swing and stance foot (red and green curves, respectively) have been plotted relative to the first step. Note that the scale of coherence level for all subjects is half the scale presented in coherence graphs for the individual subject.

DOI: https://doi.org/10.7554/eLife.36123.004

termination of locomotion. Coherence, however, was non-significant for the last double support phase prior to the transition in most subjects (n = 8), even though participants performed a complete final step, which displayed similar kinetics and kinematics to the previous steps (shaded grey areas, *Figure 5*).

During the transition period from locomotion to quiet standing (i.e. between the grey and red lines), EVS-GRF coherence increased past the 99% confidence limit (>0.045 for individual subjects) and remained significant into, and for the duration of, quiet standing (*Figure 5*). During the transition, on average, coherence peaked at a magnitude of 0.41 ± 0.08 and 0.24 ± 0.08 at frequencies of 3.3 ± 0.8 and 2.4 ± 0.4 Hz for the head forward and the head left conditions (*Figure 3*), respectively.

Subjects exhibited larger EVS-GRFs coherence during quiet standing at the end of each trial and during the transition than during the last step (refer to *Figure 1* for the total duration representing the final step), regardless of head orientation. On average, with the head facing forward, peak coherence during the last step was significantly smaller than during the transition ($\Delta = -0.27 \pm 0.09$, $t_{(9)} = -9.22$, p<0.001) and during quiet standing ($\Delta = -0.29 \pm 0.1$, $t_{(9)} = -9.02$, p<0.001). Similar results were observed when the head was turned to the left (last step versus transition: $\Delta = -0.15 \pm 0.09$, $t_{(9)} = -4.65$, p=0.001; last step versus quiet standing: $\Delta = -0.33 \pm 0.08$, $t_{(9)} = -12.64$, p<0.001). Peak coherence was not different between quiet standing and transition periods in the head forward condition (p=0.857), but was significantly higher during quiet standing when compared to the transition period with the head turned to the left ($\Delta = 0.18 \pm 0.09$, $t_{(9)} = -6.21$, p<0.001) (*Table 1* and *Figure 3*).

Finally, the kinematics and kinetics of the quiet standing period at the end of each trial were similar to those observed during the quiet standing period at the start of each trial (i.e. small amplitude oscillations at low frequency). Similarly, significant EVS-GRF coherence persisted for the duration of the quiet standing period at the end of trials, with coherence peaking at a magnitude of 0.43 ± 0.09 at 3.1 ± 0.9 Hz with the head forward and 0.42 ± 0.07 at 2.0 ± 0.5 Hz with the head left (*Figure 3*).

## A vestibulo-motor null period also preceding the transition between two standing postures

During the quiet standing period preceding the shift in postures, subjects swayed with low amplitude horizontal linear acceleration (<0.02 ± 0.2 m/s$^2$), generating small GRF (root-mean-square of 0.44 ± 0.4 N). All subjects (n = 6) exhibited significant EVS-GRF coherence that peaked with a magnitude of 0.45 ± 0.21 at 3.6 ± 1.2 Hz (*Figure 6*).

Before the onset of the posture-to-posture transition (vertical red line, *Figure 6*), coherence decreased in all subjects (n = 6), while the linear acceleration of the head and the angular velocity of the head both remained in the range observed during the quiet standing period. On average, coherence fell below the 99% confidence limit 0.336 ± 0.230 s before the onset of the transition and remained below significant level for 0.810 ± 0.351 s. Coherence returned during the transition, and remained significant into, and for the duration of, the new standing posture period.

On average, coherence peaked at a magnitude of 0.53 ± 0.07 at 4.7 ± 1.9 Hz during the transition period (i.e. between the red and grey vertical lines, *Figure 6*), and at a magnitude of 0.51 ± 0.08 at 4.8 ± 2.3 Hz in the shifted posture period (i.e. after the vertical grey line). Statistical analysis revealed that peak coherence measured during each of the three movement phases (quiet standing posture,

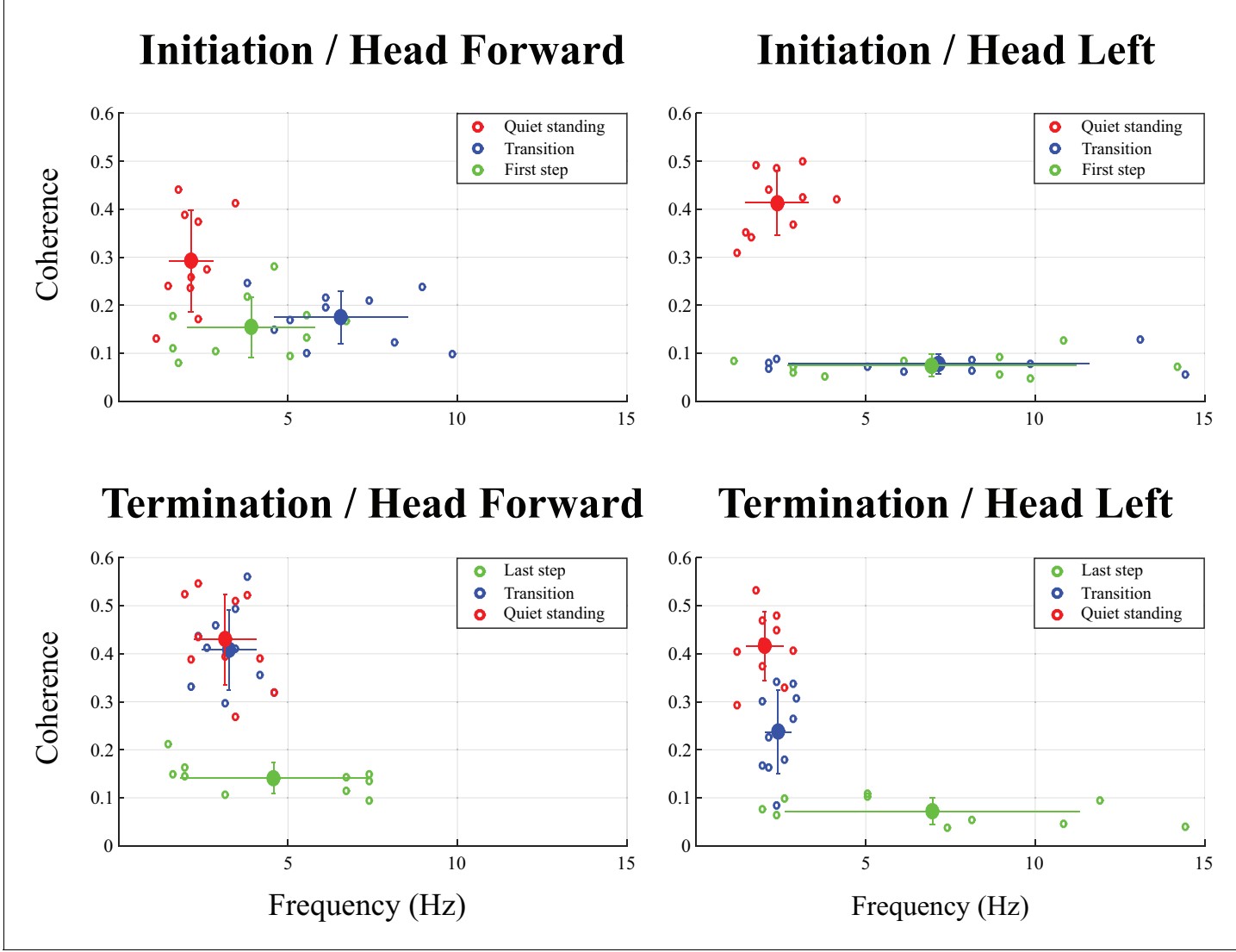

**Figure 3.** Magnitude and frequency of peak coherence measured at locomotion initiation (top) and termination (bottom) over three time periods: quiet standing (red dots), transition (blue dots) and first or last step (green dots). The big dots represent average values across all subjects (n = 10) and each small dot represents individual subject (n = 1). The results presented on the left panel are for the head forward condition; those in right panel, for the head left condition. Error bars represent one standard deviation of peak coherence magnitude (vertical) and frequency (horizontal). Multivariate analysis of variance (MANOVA) revealed a significant effect of the movement period in each of the four conditions (all p<0.001). Multivariate analysis comparing results from two periods showed the quiet standing period was significantly different from the two other periods (transition and first/last step, p<0.001), except for the head left condition when subjects terminated locomotion (p=0.857). Results of a univariate *t*-test comparing coherence peak and frequency of that peak across the different movement periods are provided in *Table 1*. The high variability in the frequency at which maximal coherence was observed in the head left condition during the transition, the first step at initiation, and the last step at termination is due to the very low level of coherence measured during these periods.

DOI: https://doi.org/10.7554/eLife.36123.005

transition, standing posture with new weight distribution) were not different from each other (F$_{(2,5)}$ = 0.76, p=0.56).

## Discussion

The main finding of this study is that the vestibular influence on balance control is interrupted prior to the onset of the transition period: ~0.26–0.44 s before transitioning from quiet standing to loco-motion or to a new standing posture, and during the last step from locomotion to standing posture.

**Table 1.** Summary of statistical tests used to compare coherence peak magnitude and its corresponding frequency for the two head positions during locomotion initiation (three time periods: quiet standing, transition and first step, top rows) and locomotion termination (three time periods: last step, transition and quiet standing, bottom rows).
Abbreviations used: HF = Head Forward; HL = Head Left; QS = Quiet Standing period; TR = transition period; FS = First Step period; LS = Last Step period; Peak = coherence peak magnitude; Freq = Frequency.

**MANOVA and *post hoc* tests results**

| | INITIATION | | QS versus TR | | | QS versus FS | | | TR versus FS | | |
|---|---|---|---|---|---|---|---|---|---|---|---|
| | Period effect | | Multi variate | Univariate | | Multi variate | Univariate | | Multi variate | Univariate | |
| | F | p-value | | Peak | Freq | | Peak | Freq | | Peak | Freq |
| HF | 14.15 | <0.001 | <0.001 | 0.009 | <0.001 | <0.001 | 0.012 | 0.015 | 0.073 | - | - |
| HL | 103.76 | <0.001 | <0.001 | <0.001 | <0.001 | <0.001 | <0.001 | 0.011 | 0.947 | - | - |
| | TERMINATION | | LS versus TR | | | LS versus QS | | | TR versus QS | | |
| | Period effect | | Multi variate | Univariate | | Multi variate | Univariate | | Multi variate | Univariate | |
| | F | p-value | | Peak | Freq | | Peak | Freq | | Peak | Freq |
| HF | 21.20 | <0.001 | <0.001 | <0.001 | 0.228 | <0.001 | <0.001 | 0.215 | 0.857 | - | - |
| HL | 30.23 | <0.001 | <0.001 | 0.001 | 0.007 | <0.001 | <0.001 | 0.007 | <0.001 | <0.001 | 0.148 |

DOI: https://doi.org/10.7554/eLife.36123.007

This suspension of the contribution of vestibular feedback to the control of balance provides critical evidence that balance correcting responses are down-regulated to allow the transition between motor states and is suggestive of the disengagement of one control policy prior to the implementation of another, supporting predictions by OFC theory (*Cluff and Scott, 2016*).

## Balance correcting responses are down regulated to enable a transition

In order to transition between two motor states, balance correcting responses are disrupted. Indeed, prior to the transitions tested, the vestibular influence on whole-body balance responses is suspended. The suspension of vestibular-evoked responses at movement initiation starts while subjects are still maintaining quiet upright stance, well before the transition onset (vertical red line, *Figures 2*, *4* and *6*). At locomotion termination, the suspension begins during the final step and ends after the transition onset (vertical grey line, *Figure 1 and 5*). Thus, changes in vestibulo-motor control occur *before* overt markers of the biomechanical transition. In addition, inhibition of ankle plantar flexor muscular activity during locomotion initiation occurs 150 ms before the first change in vertical GRF (*Herman et al., 1973*), suggesting that balance correcting responses are down regulated 100 to 300 ms before the step-related modulation in muscle activity. These results partly confirm and extend previous work that reported a decrease in vestibular influence at locomotion initiation but increased vestibular influence at locomotor termination (*Bent et al., 2002*). The apparent contradiction between these two studies is likely due to the near continuous approach we used, which improves upon the resolution with which vestibulo-motor responses can be observed compared to the discrete stimuli used by *Bent et al. (2002)*.

The period of null coherence preceding the transition was robust and observed in all individual subjects (n = 16), and it provides insight into the neural control of stabilizing mechanisms that enable motion from a stable posture. From an OFC perspective, distinct neural processes govern the control of posture and movement (*Scott, 2004*; *Kurtzer et al., 2005*; *Cluff and Scott, 2016*). To transition from quiet standing to movement, or another relatively stable posture, the current motor policy associated with quiet standing must be disengaged to permit the implementation of a new control policy (*Cluff and Scott, 2016*). In the locomotor transition task, the control policy regulating standing balance is disengaged to allow implementation of the locomotor control policy, and reversely when terminating locomotion. For the posture-to-posture transition, the motor control policy associated with standing balance is disengaged to permit the shift to a new posture. These periods of null coherence preceding transition thus illustrate *when* the brain disengages the current control policy. Furthermore, this vestibular disengagement solves the classic problem formulated by von Holst and Mittelstaedt regarding the opposing action from stabilizing reflexes when initiating motion

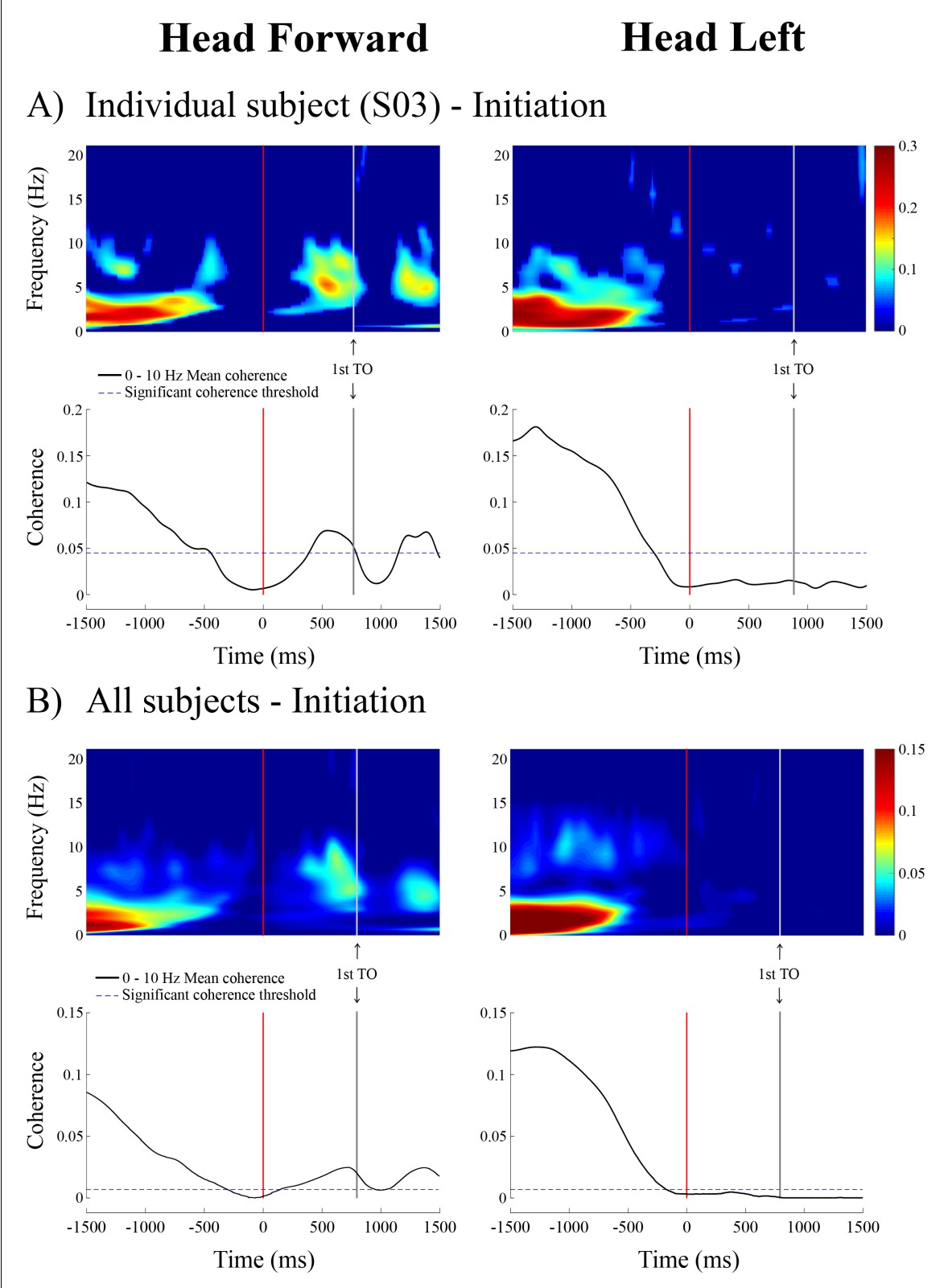

**Figure 4.** Time-frequency coherence and temporal evolution of the coherence averaged in the 0–10 Hz frequency bandwidth for a representative subject (n = 1, *panel A*, top) and averaged across all subjects (n = 10, *panel B*, bottom) at locomotion initiation in the non-normalized data segment. As in *Figure 1*, the vertical red and grey lines show the onset and the end of the transition period, respectively. Data are only presented from the 1.5 s preceding the transition onset to the 1.5 s following it (because data are less synchronized the further they are from the transition onset). In the time-

*Figure 4 continued on next page*

*Figure 4 continued*

frequency coherence graphs, all non-significant coherence values have been set to zero based on a 99% confidence limit (single subject = 0.045; pooled data = 0.005). In both A) and B), the left panel is the head forward condition and the right panel is the head left condition. Note that the scale of coherence level for all subjects is half the scale presented in coherence graphs for the individual subject. TO = toe-off.

DOI: https://doi.org/10.7554/eLife.36123.006

(*von Holst and Mittelstaedt, 1950*). From a biomechanical point of view, the disengagement of vestibular influence on the whole-body postural responses may also be necessary for the transition process, and thus incorporated into motor planning. The disruption in vestibular contribution to balance during transitions appears to contradict a key prediction from referent control theory: a monotonic shift in referent body orientation during shifts in postures while standing upright (*Mullick et al., 2018*). Our data show the intermittent use of vestibular feedback during transition between standing postures but it is unclear if a shift in referent body configuration is associated with such a transition.

The period of suspension of vestibular-evoked responses appears related to the different relative sensory feedback gains needed to control whole-body movements by the quiet standing and locomotor control policies. During quiet standing, whole-body accelerations (and related GRF) and oscillations within the limits of the base of support are small. An adaptive feedback controller (*Fitzpatrick et al., 1996*; *van der Kooij and de Vlugt, 2007*) to which vestibular feedback contributes, can generate the appropriate muscular torques required to maintain upright stance. When initiating movement, whole-body accelerations larger than those observed during quiet stance are needed to move the whole-body center of mass towards the base of support limits or outside the base of support. If the gain of the vestibular channel in the adaptive controller remained unchanged during the transition, the vestibular-evoked whole-body responses would increase due simply to the larger whole-body accelerations, potentially challenging the limits of postural stability or countering the intended motion of the body. Consequently, an adaptive controller would need to adjust the gain of the vestibular channel to maintain upright stability during this transition period. Based on our results, it appears that the transition in gains between the initial and the final policies is preceded by a suppression of vestibular gain. Vestibular gain is increased once again following the transition, but in a context-dependent manner. For example, in the event of a transition from quiet standing to locomotion, vestibular gain is increased during locomotion to control lateral stability, primarily (*Figure 2*). Similarly, it appears that the locomotor control policy is interrupted preceding locomotion termination (i.e. during the last step) to allow the implementation of the quiet standing control policy. Lastly, an alternative interpretation is that the brain abruptly switches policies and starts predicting the whole-body's future state whilst the body briefly remains in the previous state. Thus, for a short period of time, there is a discrepancy between the predicted and the actual sensory feedback that might result in a reduction in the size of the vestibular-evoked responses (*Luu et al., 2012*).

Significant coherence between EVS and the GRF were limited primarily to the 0–10 Hz bandwidth despite using EVS up to 20 Hz. These observations are in line with previous studies that characterized vestibular-evoked responses when humans are standing and walking (*Dakin et al., 2007*, *2010*). As vestibular signals cascade from the nervous system to muscle activation and from muscle activation to whole-body accelerations, the vestibular signals are low-pass filtered (*Dakin et al., 2010*). Presumably, the inertia of the limb segments and whole-body impose mechanical constraints that limit the frequencies at which their motion can remain in phase with, and thus correlate with, the EVS. These low frequency responses observed in the GRF to the vestibular stimuli are thought to reflect neural processing underlying an organised balance correcting response to a craniocentric vestibular error signal (*Reynolds, 2010*; *Mian and Day, 2014*).

Although the exact anatomical regions associated with the suppression of vestibular-evoked responses are unclear, putative regions that may contribute to control policy changes have been proposed. Primary motor cortex neurons responding to externally applied mechanical loads were found active only during the maintenance of postural orientation while others only during movement (*Kurtzer et al., 2005*; *Herter et al., 2009*). For certain cortical neurons, *Kurtzer et al. (2005)* observed an abrupt switch in the pattern of neural processing ~0.3 s before the transition from a stationary posture to a reaching movement. Our results appear to mirror this behavior and extend the framework to a balance-locomotor context. In the brainstem, selective attenuation of 'vestibular

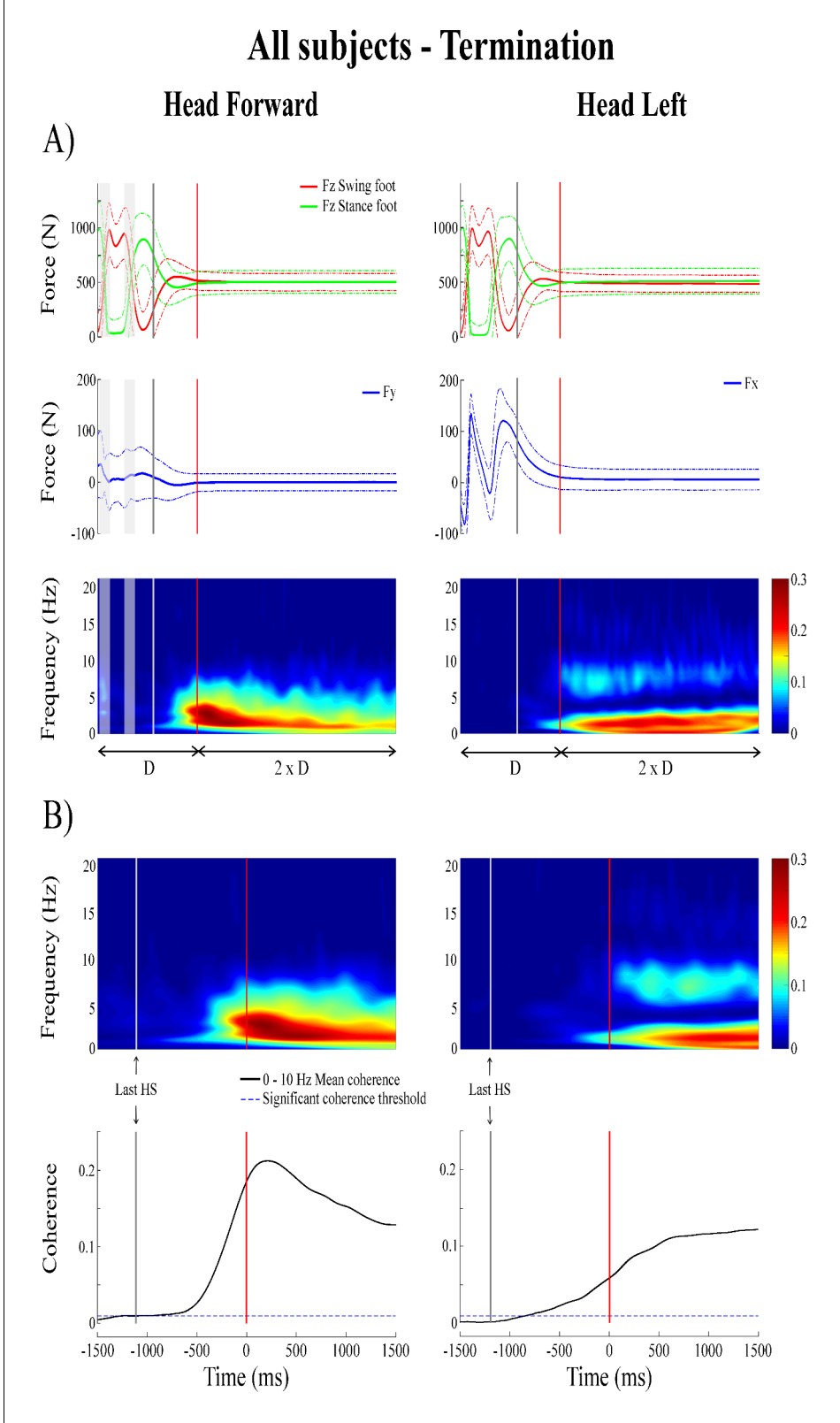

**Figure 5.** Forces and time-frequency coherence averaged across all subjects (n = 10) at locomotion termination. *Panel A* presents average results (n = 1000 trials) in the normalized time segment based on the duration (D) measured between the third-to-last foot-strike and end of the transition phase; from top to bottom: vertical reaction forces ($F_Z$), shear force in the direction of the vestibular-induced perturbation ($F_Y$ = mediolateral; $F_X$ = anteroposterior) and time-frequency coherence. *Panel B* presents averaged time-frequency coherence (top, n = 1000) and temporal evolution of the

*Figure 5 continued on next page*

*Figure 5 continued*

coherence averaged in the 0–10 Hz frequency bandwidth (bottom) in the non-normalized time segment; as in *Figure 4*, data are only presented from the 1.5 s preceding the end of the transition phase to the 1.5 s following it. In the time-frequency coherence graphs, all non-significant coherence values have been set to zero based on a 99% confidence limit equal to 0.005. In both (**A**) and (**B**), the left panel is the head forward condition and the right panel is the head left condition. In both panels, the vertical grey line shows the onset of the transition period and the vertical red line shows the end of the transition period, solid lines represent the average value and dashed line shows one standard deviation. The grey shaded areas are the double support periods during locomotion, in the head forward condition only. For illustrative purposes, the vertical forces for the swing and stance foot (red and green curves, respectively) have been plotted relative to the last step. HS = heel-strike.

DOI: https://doi.org/10.7554/eLife.36123.008

only' neurons might change the response of vestibular nuclei neurons during self-generated locomotor movement (*Roy and Cullen, 2001*). However, these neuronal responses are largely related to the predictable nature of the feedback and would therefore be ineffective at selectively suppressing an unpredictable EVS-induced perturbation signal. Although this mechanism may contribute to the attenuation of EVS-GRF responses, recent findings (*Brooks and Cullen, 2014*) indicate that it is likely insufficient to suppress the vestibular-evoked responses observed in this study.

## Active control of locomotion using vestibular feedback in the mediolateral direction

Whole-body balance responses evoked by EVS are craniocentric and thus the direction of the error can be manipulated by actively maintaining the head in different orientations (Materials and methods, *Lund and Broberg, 1983*; *Dalton et al., 2014*). In the frontal plane, our upright posture is normally unstable during locomotion because the whole-body centre of mass is always medial to the propulsive foot (*Winter, 1995*). Stability in the sagittal plane, on the other hand, is enhanced by the forward momentum of the body (*Sparrow and Tirosh, 2005*) and the long base of support during double-support phases. This difference in stability between the frontal and the sagittal planes may explain the small (or absent) vestibular-evoked responses observed during locomotion when the head is turned to the left (i.e. vestibular error directed in the anteroposterior direction). The reduced coherence between EVS and anteroposterior GRF further support the hypothesis that the anteroposterior control of whole-body balance during locomotion requires less feedback-driven control than in the frontal plane, and may instead be largely passively controlled. Thus, our results agree with the assumption that sagittal plane stability is passively controlled (*Bauby and Kuo, 2000*).

We further showed that vestibular-evoked whole-body responses in the frontal plane are phasically modulated during locomotion, exhibiting larger coherence during the double-support phase (excluding the last step at termination, *Figures 2* and *5*). Vestibulo-muscular coherence also exhibits phasic behavior for single muscles during locomotion with the head facing forward (*Blouin et al., 2011*; *Dakin et al., 2013*; *Forbes et al., 2017*), however it was not clear from summing coherence over several muscles whether the whole-body response is also phasically modulated (*Dakin et al., 2013*). From a biomechanical point of view, the double-support phase is important for stability: the base of support is larger in the mediolateral direction compared to single-support phase, the swing foot lands and body weight is transferred from one foot to another (*Winter, 1995*). Not surprisingly, EVS applied during either locomotor transition or walking leads to increased upper body roll and modified lateral foot placement (*Bent et al., 2002*). This could result in an inefficient mediolateral transfer of body weight and to steps placed more laterally than usual, raising energy consumption (*Kuo, 2002*).

## Conclusion

The disruption in the vestibular control of balance observed prior to biomechanical transition indicates that the brain first down regulates balance-correcting mechanisms to enable the transition. This disruption in the vestibular control of balance reduces counteracting effects of vestibular balance correcting responses that could prohibit movement initiation, thus supporting a prediction from current OFC models. Importantly, the suppression of vestibular balance stabilizing mechanisms is not specific to one type of transition, suggesting it represents a general control process that occurs during transitions between standing balance and other motor states. Thus, our results

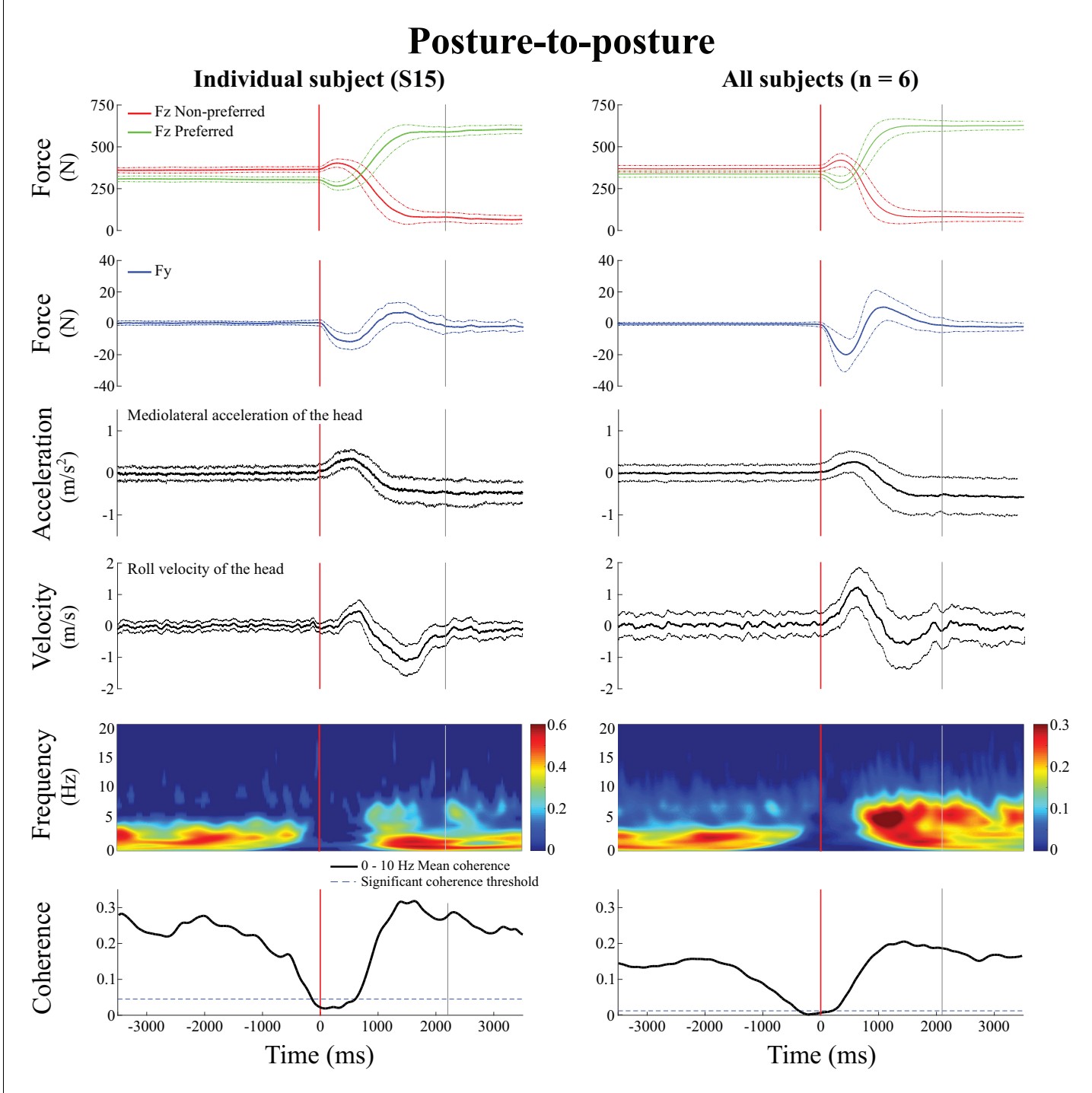

**Figure 6.** Behavioral signals and EVS-GRF time-frequency coherence during the posture-to-posture transition. Data are presented in the non-normalized time segment, that is from 3.5 s preceding the transition onset to 3.5 s following it. In both panels, from top to bottom: vertical reaction forces ($F_Z$), shear force in the direction of the vestibular-induced perturbation ($F_Y$), lateral acceleration of the head, roll velocity of the head, time-frequency coherence and temporal evolution of the coherence averaged in the 0–10 Hz frequency bandwidth. The left panel presents the results from one representative subject (S15, n = 100 trials) and the right panel presents the results averaged across all six subjects (n = 600 trials). In both panels, the vertical red line shows the onset of the transition period and the vertical grey line shows the end of the transition period, solid lines represent the average value and dashed lines show one standard deviation. In the time-frequency-coherence graphs, all non-significant coherence values have been set to zero based on a 99% confidence limit (single subject = 0.045; pooled data = 0.008). Note that the scale of coherence level for all subjects is half the scale presented in coherence graphs for the individual subject.

DOI: https://doi.org/10.7554/eLife.36123.009

demonstrate that humans 'stop balancing' before they start moving and 'stop moving' before they start balancing again.

## Materials and methods

### Population

Sixteen healthy young subjects without any history of neurological nor muscular disorders participated in our experimental protocol approved by the University of British Columbia Clinical Research Ethics Committee and conforms to the *Declaration of Helsinki*. The experimental protocols were explained to each subject and written consent was obtained. Among our participants, ten males [age 27.6 ± 4.6 years old, height 1.81 ± 0.04 m, weight 80.5 ± 9.6 kg (mean ± SD)] performed the experiment involving locomotion initiation and termination (locomotor transition) and three males and three females [age 27.5 ± 2.1 years old, height 1.74 ± 0.07 m, weight 69.9 ± 6.3 kg] performed the experiment involving a change in body-weight distribution (posture-to-posture transition).

### Stimulus

During trials, subjects were exposed to electrical vestibular stimulation (EVS) (*Dakin et al., 2007*, *2013*; *Blouin et al., 2011*) applied to the mastoid processes behind each ear. EVS non-specifically modulates the firing rate of the nearby vestibular nerves (*Goldberg et al., 1984*; *Kim and Curthoys, 2004*) and, when delivered in a binaural bipolar electrode configuration, evokes a perception of rotation (*Fitzpatrick et al., 2002*; *Peters et al., 2015*) about an axis directed posteriorly and superiorly 18° relative to the Reid's plane (*Fitzpatrick and Day, 2004*). When the head is facing forward, the binaural bipolar electrical stimulus induces a postural response to compensate for the vestibular-induced roll error signal of the body in the frontal plane (*Lund and Broberg, 1983*; *Britton et al., 1993*; *Day et al., 1997*; *Mian and Day, 2014*; *Forbes et al., 2016*, *2017*).

Subjects were exposed to a bandwidth-limited noisy EVS (0–20 Hz, zero-mean low-pass filtered white noise, 20 Hz cut-off frequency, zero lag, 4-order Butterworth, peak amplitude of 4 mA, root mean square ~2 mA) lasting 30 s (locomotor transition) or 15 s (posture-to-posture transition), and created using LabVIEW software (2011 version; National Instruments, Austin, TX, USA) (*Dakin et al., 2007*, *2013*). EVS was delivered to subjects using an isolated constant current stimulation unit (STMISOLA, Biopac Systems, Goleta, CA, USA) connected to carbon rubber electrodes (~9 cm$^2$) coated with conductive gel (Spectra 360, Parker Laboratories, Fairfield, NJ, USA) and secured over their mastoid processes with adhesive tape and an elastic headband. During the experiment, the stimulation unit remained in a torso-pack worn by the subject. The stimulus bandwidth (0–20 Hz) was chosen to characterize the entire frequency response of vestibular-induced modulation in lower limb muscle and whole-body force production (*Dakin et al., 2007*, *2010*, *2011*; *Mian et al., 2010*; *Reynolds, 2010*). The stimulus amplitude varied between ±4 mA, and was chosen to evoke measurable force responses during both standing and walking and ensure subject comfort, based on previous experience (*Dakin et al., 2007*, *2013*; *Blouin et al., 2011*; *Forbes et al., 2017*). Across all subjects (n = 16), six were new to EVS.

### Experimental design

First, subjects stood with each foot over distinct forceplates (FP1 and FP5, *Figure 1*), their arms at their side and eyes open. Subjects self-selected both the position and orientation of their feet, to ensure their comfort, but were encouraged to choose a foot width close to their pelvis width, because support width influences the magnitude of EVS-evoked responses (*Day et al., 1997*). Once chosen, the position and orientation of the feet were marked on the ground, to ensure a reproducible starting position across trials.

Subjects wore a head-mounted laser pointer and were instructed to maintain the projected position of this laser at a specific point on the wall, which was 4 m in front of them at the start of the trial. We determined the target for the laser by positioning participant's head nose up from the floor, with their Frankfurt plane (the auriculo-orbital plane) approximately 18° up from horizontal, to maximize the EVS-evoked balance responses (see *Fitzpatrick and Day, 2004*, for a review). Participants were required to maintain this head orientation for the duration of the trial. The experimenter visually monitored the inclination angle of the head during the trial.

To control the modulating effect of head orientation on the vestibular-induced instability during locomotor transition, participants completed trials in one of two head orientations. During this experiment, subjects repeated blocks of 105 trials with the head facing forward or turned 90° over the left shoulder (*Figure 1*). During the posture-to-posture experiment, subjects performed only one block of 105 trials with the head facing forward. When the head is facing forward, EVS induces a postural response along the mediolateral whole-body axis. When the head is turned 90° to the left, postural responses are directed along the anteroposterior whole-body axis (*Lund and Broberg, 1983*; *Fitzpatrick et al., 1994*; *Forbes et al., 2016*). Subjects participating in the locomotor transition experiment performed blocks of trials on separate days and we randomly assigned the order of the two blocks for each subject. During each block, the first five trials were used to collect the subject's kinetic and kinematic data in absence of EVS. These trials served as a control to determine if EVS modified the movement pattern. We provided EVS for the remaining 100 trials. Statistical analysis of the movement patterns between trials with and without EVS were performed using two-sided *t-tests* for each head position condition. In the locomotor transition experiment, the analysis did not reveal any major behavioral difference between these two conditions, except at locomotion termination, for a significantly decreased transition period ($\Delta = -0.094 \pm 0.070$ s, $t_{(54)} = -4.93$, p<0.001 and $\Delta = -0.330 \pm 0.123$ s, $t_{(51)} = -7.49$, p<0.001) and an increased walking velocity ($\Delta = 0.044 \pm 0.013$ m/s, $t_{(50)} = -15.79$, p<0.001 and $\Delta = 0.113 \pm 0.070$ m/s, $t_{(51)} = 73.82$, p<0.001), respectively in the head forward and head left conditions. In the posture-to-posture experiment, we did not find any significant behavioral differences between the trials with and without EVS.

## Locomotor transition experiment

The beginning of each trial was indicated by a short (250 ms) vibration pulse (20 Hz) applied at waist level by a vibrator (FG-142, Labworks Inc., Costa Mesa, CA, USA) secured on a waist band. During trials with EVS, EVS started simultaneously with the vibration pulse (*Figure 1*). We instructed subjects to 'stand as naturally as possible and maintain a quiet standing posture' at the start of the trial. About five seconds after the trial onset (vibration pulse), the experimenter verbally informed subjects that they could 'initiate locomotion whenever they wanted'. After this go cue, subjects could initiate locomotion with their preferred foot at their own discretion. Subjects walked straight and at their preferred speed, across the length of four forceplates, and continued until they were 1 m past the last forceplates (FP4 and FP8, *Figure 1*). At this point (~3.5 m from their starting point), subjects were instructed to stop, turn around to face the opposite direction, and, at the same speed, return toward their starting position. We chose the walking distance to allow subjects sufficient space to reach their preferred locomotion speed, which occurs on average after the second step (*Jian et al., 1993*). Subjects terminated their locomotion once they had reached the same start position with their feet (i.e. one foot on FP1 and one on FP5, *Figure 1*). We instructed subjects to stand upright, as naturally as possible, after terminating locomotion, until they received a second vibration pulse (250 ms, 20 Hz) indicating the end of the trial. During trials with EVS, EVS stopped simultaneously with the second vibration pulse. Trials lasted 30 s and therefore the two vibration pulses occurred 30 s apart. In total, each trial consisted of at least five seconds of quiet standing data before the initiation of locomotion, locomotion for 3.5 m forward and back, and at least five seconds of quiet standing data following the termination of locomotion. During this experiment, a trial was considered invalid if the subjects (i) did not use the same foot to initiate locomotion, (ii) did not use the same foot to terminate locomotion, (iii) terminated locomotion with part of the feet outside of the forceplates or (iv) changed their head position before the end of the trial while being on the forceplates. In total, only 50 out of 2100 trials performed were considered invalid (2.3%). To account for this possibility and ensure each subject completed the 100 trials necessary for the analysis, we asked subjects to complete ten additional walking with EVS trials (110 trials with EVS total).

## Posture-to-posture experiment

The experimental protocol was similar to the locomotor transition experiment. Briefly, we instructed subjects to maintain equal weight distribution beneath their feet and then shift their weight laterally, in order to move 90% of their whole-body weight towards their preferred leg, without changing their feet position. A vibration pulse (20 Hz, 250 ms) applied to the subjects' waist indicated trial initiation. At least five seconds following this vibration pulse, subjects were verbally informed that they could

'shift their weight whenever they wanted'. Following this verbal start cue, subjects shifted their weight to their preferred leg with a 90–10% weight distribution, and then they were asked to maintain this position until a second vibration pulse (20 Hz, 250 ms) indicated the end of the trial and corresponded to the end of the EVS stimulus. The two vibrations pulses occurred 15 s apart and therefore each trial lasted 15 s. Weight distribution underneath the feet was monitored by an experimenter at all times. During this experiment, a trial was considered invalid if the subjects (i) did not shift their weight to their preferred leg, (ii) stabilized themselves with a load on the preferred leg more or less than the preferred range of 85–95% of body weight, or (iii) changed their head position before the end of the trial. In total, only 6 out of 600 trials performed were considered invalid (1%). We asked subjects to complete five additional trials with EVS (105 trials total) to account for this possibility and ensure each subject completed the 100 trials necessary for the analysis.

## Instrumentation

To quantify the effect of the vestibular stimulus on subjects' behaviour, we recorded the ground reaction forces (GRF) applied to their body as well as linear acceleration and angular velocity of their head over the duration of each trial. The GRF were collected from eight forceplates (squares of $60 \times 60$ cm, AMTI, Watertown, MA, USA) recessed into the ground of the experimental room, covering an area of 2.88 m$^2$. GRF were digitized using a sampling rate of 1000 Hz (PXI-8108, National Instruments, Austin, TX, USA) and saved offline, with the vestibular stimulus and the vibration pulse signals.

We measured the linear acceleration and angular velocity of the head and waist using two inertial measurement units (Shimmer2R w450/mAH Battery, $5.1 \times 3.4 \times 1.4$ cm, Shimmer Sensing, Dublin, Ireland) positioned on the forehead and the sacrum. The inertial measurement units were secured to the head and waist using Velcro belts. During trials, data from the inertial measurement units were sampled at 256 Hz and streamed to a personal computer for storage and future analysis. Because we used two separate computers to save the forceplate and inertial measurement units' data, the two were synchronized in post-processing using the onset of the first vibration pulse.

## Data reduction and signal analysis

After identifying the biomechanical markers of the transitions phases, we analyzed the vestibular responses during the transition between standing balance and locomotion and between the two standing postures to determine if vestibular feedback is disrupted as predicted by the OFC theory. All non-statistical analyses were performed using custom-designed routines on Matlab software (2015a version, Mathworks, Natick, MA, USA).

## Identification of the movement phases

### Locomotor transition

We identified toe-offs and heel-strikes in each trial by using the vertical component of the GRF (**Figure 1**). Toe-offs were defined as the first vertical force data point in which the swing foot was less than five percent of the subject's body weight, and heel-strikes were defined as the first vertical force data point after a toe-off where the swing foot was greater than five percent of the subject's body weight. Following identification of the toe-offs and heel-strikes, the data were cut from first vibration pulse to the third heel-strike (segment of data for initiation trials) and from the third-to-last toe-off to second vibration pulse (segment of data for termination trials). In the cut data, the absolute value of the horizontal velocity of the centre of pressure was computed from the first time derivative of its trajectory, after being filtered with a zero-phase, second-order Butterworth digital filter at a 20 Hz cut-off frequency. Using this variable, the onset of the transition phase at locomotion initiation and the end of the transition phase at locomotion termination were identified as the first and last instants when the horizontal velocity of the centre of pressure was greater than three standard deviations from baseline. Baseline was defined as the first five seconds of the quiet standing period at the onset of each trial, and the last five seconds of the quiet standing period at the end of each trial. The end of the transition phase between quiet standing and locomotion at locomotion initiation was identified as the first toe-off (**Figure 1**). The onset of the transition phase between locomotion and quiet standing at locomotion termination was identified as the last heel-strike. The transition from quiet standing to locomotion (between the red and grey lines, **Figure 1**) was characterized by

a loading (increase in vertical force resulting in movement of the centre of pressure toward the swing leg and increase in shear force moving the CoM toward the stance leg) and then unloading of the swing leg (decrease of vertical force moving the centre of pressure away from the swing leg and reversed increase in the shear force to slow the center-of-mass's fall toward the stance leg).

### Posture-to-posture transition

We used the first instant when the horizontal velocity of the centre of pressure was greater than three standard deviations from baseline to determine the onset of the transition period between the two standing postures. Baseline was defined as the first five seconds of the quiet standing period at the onset of each trial. This transition is similar to locomotion initiation, as a loading-unloading mechanism is used to shift balance between postures.

For all trials, the onset and end of the transition period was identified automatically by a Matlab routine and visually verified by an experimenter. In the event of a misidentification, values were manually corrected (~15% of the time).

## Vestibular responses

The relationship between EVS and the GRF for each experiment was estimated using time-frequency coherence based on continuous Morlet wavelet decomposition (*Zhan et al., 2006*; *Blouin et al., 2011*). This method has been previously used to estimate the changes in vestibular-evoked responses measured by EMG (*Blouin et al., 2011*; *Luu et al., 2012*; *Dakin et al., 2013*; *Forbes et al., 2014*, *2017*) and was chosen here to avoid potential issues associated with gain estimates due to the larger GRF during locomotion (and transitions) than during quiet standing. We evaluated the coherence between the vestibular stimulus and the horizontal GRF because they reflect the net muscle activity of the body and are related to the horizontal accelerations of the whole-body centre of mass. Coherence [$C(\tau,f)$] used to estimate vestibulo-GRF coupling was computed using the following equation:

$$C(\tau,f) = \frac{|P_{xy}(\tau,f)|^2}{P_{xx}(\tau,f)\, P_{yy}(\tau,f)} \tag{1}$$

In *Equation (1)*: $\tau$ is the movement time; $f$ denotes the frequency; $P_{xy}(\tau,f)$ is the time-dependent cross-spectrum between the EVS and the shear force of interest; and $P_{xx}(\tau,f)$ and $P_{yy}(\tau,f)$ are the time-dependent auto-spectra of the EVS and the shear force of interest, which depends on head orientation. When the head faces forward, the perturbation resulting from the vestibular stimulus is directed mediolaterally, therefore the mediolateral shear forces ($F_Y$) were used for the analysis of the head forward condition. When the head is turned to the side, the perturbation resulting from the vestibular stimulus is in the anteroposterior direction, therefore the anteroposterior shear forces ($F_X$) were used for the head left condition.

### Locomotor transition

In trials with EVS, we extracted 100 data segments per subject for locomotion initiation and termination for both head orientations (400 segments in total) from the cut data described previously (see previous paragraph). Each segment was saved in two separate time-scale formats: movement-time normalized and non-normalized. In the movement-time normalized format, we stretched or compressed the length of the movement time ($\tau$) to account for variability in the timings of toe-off, heel-strike and step duration, within each trial, between trials, and between subjects. To normalize $\tau$ in each segment at locomotion initiation, we identified the time from the onset of the transition to the third toe-off (duration D illustrated in *Figure 2*). Then, we extracted that data, with additional data from the quiet standing period prior to the transition of twice the length of the transition onset to third toe-off period. Similarly, to normalize $\tau$ for each segment at locomotion termination, we extracted data from the third-to-last toe-off to the end of the transition (duration D illustrated in *Figure 5A*) with data of twice that length from the quiet standing period immediately following the end of the transition period. We decomposed the data on a trial-by-trial basis using a wavelet transformation, with a frequency resolution of 1 Hz and limited to the 0-20 Hz bandwidth, and with a lag time of 200 ms to maximize the correlation associated with the vestibular correcting responses (often referred to as the medium latency response [*Dalton et al., 2014*; *Mian and Day, 2014*]). To confirm

this choice, we explored the EVS-GRFs coherence using lag times in the 200-500 ms range (covering the duration of the vestibular balance correcting response) and observed minimal effect of these lags on the coherence results. We then interpolated or down-sampled the auto-spectra and cross-spectrum of these segments to a common length, their average length across all trials (repetitions, conditions and subjects) (*Blouin et al., 2011*). For each head orientation and locomotor transition combination, we calculated the coherence spectrum between EVS and the GRF in each subject (100 segments) as well as on data pooled across all subjects (1000 segments). In order to characterize the changes in the vestibular-evoked balance responses over an interpretable time scale (s), we also performed the analysis without stretching or compressing $\tau$ (i.e. the non-normalized time-scale format). We cut the data from both initiation and termination of locomotion to place the onset of the transition period at initiation and the end of the transition period at termination in the middle of a three-second data window. We then estimated the EVS-GRF coherence spectrum on this un-normalized data using the same procedure described previously. This analysis was performed both on the data for each subject (100 segments) as well as on the data pooled across all subjects (1000 segments) for each head orientation and locomotor transition combination. To cancel the shift in coherence timing induced by the 200 ms lag used in the time-frequency analyses, results and figures are presented with a 200 ms time-shift in the non-normalized time-scale analyses and an equivalent 200 ms time-shift in the normalized time-scale analyses.

## Posture-to-posture transition

We extracted 100 data segments per subject, which we analyzed using only the non-normalized time format. For these analyses, the onset of the transition was placed in the middle of a 7 s window. Similar to the locomotor transition analyses, data were decomposed on a trial-by-trial basis using a wavelet transformation. Coherence between EVS and the GRF was then calculated for each subject (100 segments) as well as on data pooled across all subjects (600 segments).

## Statistical analysis

Significant whole-body responses to the EVS were identified on a subject-by-subject basis when coherence exceeded a 99% confidence limit determined by the number of trials completed (n = 100). This confidence limit corresponds to a coherence magnitude threshold of 0.045, and better represents an $\alpha$-level of 0.05 because of the two-dimensional (time and frequency) nature of the measures (*Blouin et al., 2011*). For illustrative purposes, we also computed the coherence for all subjects using the same procedure (n = 1000 for the locomotor transition or n = 600 for the posture-to-posture transition). Coherence crossed the 99% confidence limit in these pooled data at a magnitude of 0.005 (locomotor transition) or 0.008 (posture-to-posture transition). Under OFC theory, we expected a discontinuity in time-frequency coherence prior to the different transitions which should be reflective of the disengagement of either the balance, or (loco)motor control policies. To test this hypothesis, using the non-normalized time-scale format, we determined whether the magnitude of time-frequency coherence was significant or not (i.e. above or below the 99% confidence interval, respectively) over the transition period. To identify state-dependent changes in vestibular control of balance over the locomotor transition period, the magnitude of the peak time-dependent coherence and its corresponding frequency were extracted from the movement-time normalized data in each subject and for the three phases of the balance-locomotor transitions: quiet standing (from first vibration pulse to transition onset for initiation and from end of transition to last vibration pulse for termination), transition (from transition onset to first foot-off for initiation and from last foot-strike to transition end for termination) and the first/last step (from first foot-off to second foot-off for initiation and from second last foot-strike to last foot-strike for termination) (*Figure 1*). The same procedure was repeated for the posture-to-posture transition (but for the non-normalized time format) for the three following steps: quiet standing (from first vibration pulse to transition onset, i.e. vertical red line), transition (from transition onset to transition end, i.e. between the two vertical lines) and the new posture (from transition end, i.e. the vertical grey line, to the end of the trial) (*Figure 6*). Peak magnitude and/or frequency were used to determine whether coherence changed between the three phases, relative to the direction of vestibular-evoked responses. To test for significant differences across the three phases of each transition tested, we used repeated-measures multivariate analysis of variance (MANOVA). To decompose the main effect, we used a multivariate *post hoc*

analysis consisting of paired Hotelling's $t$-square tests (Bonferroni corrected). For each multivariate *post hoc* test providing a significant result, we tested whether the significant result was caused by differences in magnitude peak coherence, peak coherence frequency, or both by performing a Bonferroni corrected paired $t$-test. In the locomotor transition, this whole procedure was repeated independently for each head orientation within each transition (initiation and termination). Coherence confidence limits were computed using Matlab software (2015a version, Mathworks, Natick, MA, USA). All other statistical tests were performed using R Studio software (version 3.3.1) with an $\alpha$ level of $p<0.05$ serving as the threshold for significance.

## Acknowledgments

Authors thank Cole Shin for technical support and Dr. David Bolton, Dr. Patrick Forbes and Dr. Aaron Palmer for relevant comments on the manuscript. This study was funded by the Natural Sciences and Engineering Research Council of Canada, grant number 356026–13 (J-S B).

## Additional information

### Funding

| Funder | Grant reference number | Author |
|---|---|---|
| Natural Sciences and Engineering Research Council of Canada | 356026–13 | Jean-Sébastien Blouin |

The funders had no role in study design, data collection and interpretation, or the decision to submit the work for publication.

### Author contributions

Romain Tisserand, Conceptualization, Data curation, Software, Formal analysis, Validation, Investigation, Visualization, Methodology, Writing—original draft, Project administration, Writing—review and editing; Christopher J Dakin, Conceptualization, Formal analysis, Supervision, Validation, Visualization, Methodology, Writing—original draft, Writing—review and editing; Machiel HF Van der Loos, Elizabeth A Croft, Conceptualization, Resources, Writing—review and editing; Timothy J Inglis, Conceptualization, Visualization, Writing—review and editing; Jean-Sébastien Blouin, Conceptualization, Resources, Software, Formal analysis, Supervision, Funding acquisition, Validation, Investigation, Visualization, Methodology, Writing—original draft, Project administration, Writing—review and editing

### Author ORCIDs

Romain Tisserand https://orcid.org/0000-0002-6857-9886
Christopher J Dakin http://orcid.org/0000-0002-6781-0281

### Ethics

Human subjects: Informed consent of participating to this study and publishing the results in a scientific journal was obtained from all participants. Ethical approval was obtained from the University of British Columbia Clinical Research Ethics under the identifiant H09-00987. Application was approved by Suzanne Richardson.

### Decision letter and Author response

Decision letter https://doi.org/10.7554/eLife.36123.016
Author response https://doi.org/10.7554/eLife.36123.017

## Additional files

### Supplementary files

• Source code 1. MANOVA and Hotelling's t-square test in R.
DOI: https://doi.org/10.7554/eLife.36123.010

• Source code 2. Time frequency Coherence Analysis script in MATlab (Calls Morlet Wavelet Transform Functino and Total Coherence Function).
DOI: https://doi.org/10.7554/eLife.36123.011

• Source code 3. Morlet Wavelet Transform Function in MATlab.
DOI: https://doi.org/10.7554/eLife.36123.012

• Source code 4. Total Coherence Function in MATlab.
DOI: https://doi.org/10.7554/eLife.36123.013

• Transparent reporting form
DOI: https://doi.org/10.7554/eLife.36123.014

### Data availability

The time-frequency coherence analysis codes (Matlab) and the statistical analysis code (R) are provided and the anonymized human data are maintained on a University of British Columbia file server. This data sharing interface requires that the link to the dataset be changed regularly for security reasons. As such, a current link to the data is available through a request to Jean-Sébastien Blouin at jsblouin@mail.ubc.ca

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
