## [Decision Letter]

Thank you for submitting your article "Transitions between posture and locomotion involve down regulation of balance stabilizing mechanisms" for consideration by *eLife*. Your article has been reviewed by three peer reviewers, including Richard Staines as the Reviewing Editor and Reviewer #1, and the evaluation has been overseen by Richard Ivry as the Senior Editor.

The reviewers have discussed the reviews with one another and the Reviewing Editor has drafted this decision to help you prepare a revised submission.

Summary:

This paper investigates the neural control of transitions between posture and movement. The interaction of vestibular information and balance-correcting mechanisms was investigated during transitions between quiet standing and locomotion. Electrical vestibular stimuli (EVS) were applied during stance and locomotion and the relationship between the vestibular stimuli and ground reaction forces were assessed using a time-frequency analysis as coherence between the two. The results show a clear low frequency coupling between EVS and ground reaction force during standing and a phase-dependent coupling observed only during the stance-phase of locomotion. The coupling disappeared approximately 200-300 ms before the transition phase from standing to the onset of locomotion, and rotating the head 90 degrees switched the coupling from mediolateral to frontoposterior directions, as expected based on the directional influence of EVS. The experimental design was developed to compare and contrast predictions from two general theories of motor control, Optimal feedback control (OFC) and equilibrium models. Overall, the experimental design is clever and highlights clear changes in the motor response to vestibular input from quiet stance to the transition of locomotion, phasic use of vestibular input during locomotion and changes again from locomotion to quiet stance. The data provides compelling support that feedback is context-dependent, suggesting control policies abruptly change as one switches between different types of behaviors, as expected within the OFC framework. Interestingly, the fact that the phase-dependent vestibular influence stops early before the last step suggests the last step may be kinematically similar to previous steps but the underlying control is quite distinct.

Essential revisions:

1) The primary concern is the lack of a control experiment that differentiates between transitions to pure walking (as tested) and adjustment for expectation of a new posture. One argument is that during the transition periods, the control system might be preparing for new posture before the walking actually begins. Specifically, it is unclear whether the observed change in the coherence is specific to the transition between posture to walking or/and it is more general control process of changes between different states (posture<->new posture or posture<->walking). A possible control for this would be testing participants while they switch between two/more postures that doesn't require walking (i.e., a single step to a new posture or simply a shift of weight largely from one leg to the other while maintaining constant balance). The main point here is the demonstration of this in a different transition to strengthen and generalize the important conclusion of this manuscript.

2) The experiments in this manuscript show that vestibular feedback is task dependent and turns off before the transition to initiate gait, is phasically involved in control during locomotion (only during double stance), and turned off for the last step before movement. Although this is good evidence that the entire motor system appears to be changing control policies, the degree to which the data is evidence against equilibrium models is less clear. Those that espouse the equilibrium-point framework may argue that the referent body configuration is maintained from quiet standing to locomotion and back again, and what this data shows is intermittent use of vestibular feedback. Unless you can find some argument about the continuous influence of non-proprioceptive feedback for the equilibrium point framework, you should probably acknowledge this alternative interpretation in the discussion and alter the introduction accordingly. It is suggested that the authors downplay the equilibrium theory and focus on flexible feedback processing with a focus on the novelty of task-dependent vestibular feedback processing related to posture, locomotion and transitions.

3) The quantitative description of the time-frequency analysis for the coherence measure is well described from previous work from this group and the results clear. However, more discussion as to the importance of this interaction between EVS-GRF in the frequency range of 0-10 Hz reflected by the coherence would be helpful. Specifically, a brief description of the neural mechanisms that contribute to this measure should be included.

---

## [Author Response]

Essential revisions:1) The primary concern is the lack of a control experiment that differentiates between transitions to pure walking (as tested) and adjustment for expectation of a new posture. One argument is that during the transition periods, the control system might be preparing for new posture before the walking actually begins. Specifically, it is unclear whether the observed change in the coherence is specific to the transition between posture to walking or/and it is more general control process of changes between different states (posture<->new posture or posture<->walking). A possible control for this would be testing participants while they switch between two/more postures that doesn't require walking (i.e., a single step to a new posture or simply a shift of weight largely from one leg to the other while maintaining constant balance). The main point here is the demonstration of this in a different transition to strengthen and generalize the important conclusion of this manuscript.

We thank the reviewers for this comment. We performed a control experiment, using the same protocol as we did for locomotion but, according to the reviewers’ recommendation, we asked participants to shift their weight from 50% on each leg to 90% on their preferred leg without lifting the feet at any time. Because this movement involves a lateral displacement, we tested participants only in the head forward condition (EVS perturbation directed mediolaterally). The results from this experiment clearly show that the vestibular stabilizing mechanisms are interrupted when participants initiate the movement necessary to adopt a new posture. The duration of this vestibular suspension (810 ms) was similar to that observed during locomotion initiation (860 ms). This control experiment strengthens our initial results about locomotion and suggests, as proposed by the reviewers and editor, that the change in vestibular control of balance is a general control process that applies to transitions between two motor states (posture-to-posture and posture to movement). These complementary results are presented in Figure 6. According to the new results from the control experiment, several changes have been made in the new manuscript:

Title: the title has been changed to “Down regulation of vestibular balance stabilizing mechanisms to enable transition between motor states”

Abstract: It is now stated that we performed two experiments: “we investigated the continuity of the vestibular control of balance during transitions between quiet standing and locomotion and between two standing postures” and “Healthy subjects initiated and terminated locomotion or shifted the distribution of their weight between their feet, while exposed to electrical vestibular stimuli (EVS)”.

Introduction: we now refer to our experiment as testing the vestibular control of balance “between (two) motor states” instead of the previous statement which was “between posture and locomotion” and we indicated that we “examined the contribution of vestibular sensory signals to the control of balance during transition between quiet standing and locomotion as well as between two standing postures”.

Results: An addition was made to the first paragraph to explain the second experiment (posture-to-posture): “In the second study, six additional subjects (young adults, 3 females) completed one block of 105 trials where they were instructed to stand quietly, then shift their body-weight laterally to redistribute their weight so that 90% was supported by their preferred leg and only 10% by their non-preferred leg (defined as the ‘posture-to-posture transition’ experiment). Once participants redistributed their body-weight, they were asked to maintain this uneven weight distribution. In this task, participants received EVS for the duration of each trial (15 s)”. Moreover, a new paragraph, entitled “A vestibulo-motor null period also preceding the transition between two standing postures”, has been added to describe the new results from the posture-to-posture experiment.

Discussion: As in the Introduction section, we now refer mostly to our results as a transition between “two motor states” and references to the new experiment have been added (e.g. “To transition from quiet standing to movement, or another relatively stable posture” or “For the posture-to-posture transition, the motor control policy associated with standing balance is disengaged to permit the shift to a new posture”.

Conclusion: the conclusion has been updated to include the results from the posture-to-posture experiment and states, as suggested by the reviewers and the editors, that the change in vestibular control of balance is a general control process that applies to transitions between two motor states: “Importantly, the suppression of vestibular balance stabilizing mechanisms is not specific to one type of transition, suggesting it represents a general control process that occurs during transitions between standing balance and other motor states”.

Materials and methods: the “Population” section has been updated to include the 6 new subjects who participated to the posture-to-posture experiment, which bring the total number of participants to 16 subjects. In both the “Experimental design”and “Data reduction and signal analysis”sections, the text has been changed to include one subsection describing what was done for the locomotor transition experiment and one subsection describing what was done for the posture-to-posture transition experiment.

Finally, the “Statistical analysis” section has been updated to report the analysis which have been performed on the results from the posture-to-posture experiment: “The same procedure was repeated for the posture-to-posture transition (but for the non-normalized time format) for the three following steps: quiet standing (from first vibration pulse to transition onset, i.e. vertical red line), transition (from transition onset to transition end, i.e. between the two vertical lines) and the new posture (from transition end, i.e. the vertical grey line, to the end of the trial) (Figure 6)”.

2) The experiments in this manuscript show that vestibular feedback is task dependent and turns off before the transition to initiate gait, is phasically involved in control during locomotion (only during double stance), and turned off for the last step before movement. Although this is good evidence that the entire motor system appears to be changing control policies, the degree to which the data is evidence against equilibrium models is less clear. Those that espouse the equilibrium-point framework may argue that the referent body configuration is maintained from quiet standing to locomotion and back again, and what this data shows is intermittent use of vestibular feedback. Unless you can find some argument about the continuous influence of non-proprioceptive feedback for the equilibrium point framework, you should probably acknowledge this alternative interpretation in the discussion and alter the introduction accordingly. It is suggested that the authors downplay the equilibrium theory and focus on flexible feedback processing with a focus on the novelty of task-dependent vestibular feedback processing related to posture, locomotion and transitions.

We agree with the reviewers’ recommendation about the equilibrium-point framework. Thus, we modified the text in the Introduction and Discussion sections to downplay the equilibrium theory and focus more on the processing of vestibular feedback related to transition from posture to movement (locomotion) or posture. In the Discussion we point out that the vestibular null period appears to contradict a monotonic transition in the referent body configuration but that it does not necessarily conflict with the idea of a referent body configuration driving the transition.

Introduction: “In this respect, OFC seems to diverge from other motor control theories, such as referent (threshold) control theory (Asatryan and Feldman 1965; Ostry and Feldman 2003). This latter theory suggests that a transition between postures involves a monotonic shift in the referent body orientation (Feldman et al., 2011; Mullick et al., 2018), transforming posture-stabilizing mechanisms into movement-inducing ones.”

Discussion: "The disruption in vestibular contribution to balance during transitions appears to contradict a key prediction from referent control theory: a monotonic shift in referent body orientation during shifts in postures while standing upright (Mullick et al., 2018). Our data show the intermittent use of vestibular feedback during transition between standing postures but it is unclear if a shift in referent body configuration is associated with such a transition”.

3) The quantitative description of the time-frequency analysis for the coherence measure is well described from previous work from this group and the results clear. However, more discussion as to the importance of this interaction between EVS-GRF in the frequency range of 0-10 Hz reflected by the coherence would be helpful. Specifically, a brief description of the neural mechanisms that contribute to this measure should be included.

Following the reviewers and editors’ point, we added one paragraph in the Discussion section to clarify what the interaction between EVS and GRF reflect in the 0-10 Hz frequency range: “Significant coherence between EVS and the GRF were limited primarily to the 0 – 10 Hz bandwidth despite using EVS up to 20 Hz. […] These low frequency responses observed in the GRF to the vestibular stimuli are thought to reflect neural processing underlying an organised balance correcting response to a craniocentric vestibular error signal (Reynolds 2010; Mian and Day 2014).”